# Exceptionally cold water days in the southern Taiwan Strait: their predictability and relation to La Niña

Yu-Hsin Cheng[1, 2], Ming-Huei Chang[1]

[1]Institute of Oceanography, National Taiwan University, Taipei, 10617, Taiwan.
[2]State Key Laboratory of Marine Environmental Science, College of Ocean and Earth Sciences, Xiamen University, Xiamen, Fujian, China.

*Correspondence to*: Ming-Huei Chang (minghueichang@ntu.edu.tw)

**Abstract.** The objectives of this study were to assess the predictability of exceptionally cold water in the Taiwan Strait (TS) and to develop a warning system on the base of scientific mechanism, which is a component of the information technology
system currently under development in Taiwan to protect aquaculture against extreme hazards. Optimum interpolation sea surface temperature (SST) data were used to find exceptionally cold water days from January 1995 to May 2017. We found that the SST and wind speed over the TS are low and strong in La Niña winters, respectively. According to tests conducted using relative operating characteristic curves, predictions based on the Oceanic Niño Index and integrated wind speed can be employed at lead times of 60–120 and 0–25 days, respectively. This study utilized these two proxies to develop a possible
warning mechanism and concluded four colors of warning light: (1) blue, meaning normal (0% occurrence probability); (2) cyan, meaning warning (~50% occurrence probability); (3) yellow, meaning moderate risk (~60% occurrence probability); and (4) red, meaning high risk (~75% occurrence probability). Hindcasting winters over the period 1995–2017 successfully predicted the cold water hazards in the winters of 2000, 2008, and 2011 prior to the coldest day ~20 days.

## 1 Introduction

The Taiwan Strait (TS) is a northeast-to-southwest passage, with a length of 300 km and width of 180 km, from the East China Sea to the South China Sea. The average depth of TS is 50 m and two major shallow water regions, Taiwan Bank and Chang-Yuen Ridge, are about 30 m (Fig. 1). Circulation in the TS, exhibiting strong seasonal variation, is mainly dominated by monsoon forcing and topography (Jan et al. 2002). The China Coastal Current brings cold and brackish water into the northern TS during winter from December to the following January (Jan et al. 2006; Chen et al. 2016). In addition, the strong northeast
monsoon reduces the northward transport of the Kuroshio Branch Current, bringing warm and saline water from the western North Pacific. In summer, the southwestern monsoon replaces the northeast monsoon and dominates the circulation in the TS. During the southwest monsoon season, the northward transport is intensified and brings South China Sea water into the TS (Jan et al. 2006).

El Niño–Southern Oscillation (ENSO), which develops in the tropical Pacific and is caused by the mediation between surface
wind stress and sea surface temperature (SST) variations (McPhaden et al. 2006), is an interannual climate fluctuation.

Although ENSO originates in the tropical Pacific, it significantly influences patterns of weather variability worldwide, shifting the probability for droughts, floods, heat waves, severe storms and extreme events (e.g., Alexander and Scott, 2002; Philippon et al., 2012). It is known the cold phase of ENSO, La Niña, tends to intensify the East Asian winter monsoon, which often accompanies strong northerly winds and sharp air temperature drops. By contrast, the warm phase, El Niño, suppresses the East Asian winter monsoon (Wang et al., 2000; Lau et al., 2006). In addition, Kuo and Ho (2004) indicated that the stronger northeast monsoon during a La Niña winter may modulate the sea surface currents in the TS and further cause the lower SST. For example, the southward water transport was larger and the water in the TS was generally colder in November 2000 (La Niña winter) than that in 2002 (El Niño winter). The predominance of the cold China Coastal Current and the weakness of the warm Kuroshio Branch Current resulted in the water temperature of the TS decreasing in the early winter of 2000 (Wu et al. 2007). By contrast, when El Niño broke out in the winter seasons from 1997 to 1998, the warm water area (2°C above the regional mean) in the TS increased by 25% and nutrient concentrations decreased (Shang et al. 2005). Zhang et al. (2015) suggested that less Kuroshio water enters the southeastern TS during La Niña than El Niño events, which might modulate the interannual variability of SST in the TS. However, less severe but longer lasting phenomena, such as SST variability, may have catastrophic consequences or can be a trigger for other threats (Ustrnul et al. 2015). For example, the extreme cold or hot temperature can threaten life on earth and may even trigger a disaster (Mora and Ospina 2002). Previous studies have identified a correlation between local wind and circulation in the TS, and they have suggested that a La Niña winter would be a weather condition supporting cold-event occurrences, which are more likely to trigger cold disasters in the TS.

In the winter of 2008 (December 2007 to February 2008), exceptionally cold water affect the southern TS and hit the marine natural resources around the Penghu Islands in the southwestern TS, causing damage to the aquaculture and high fish mortality rates; this phenomenon is referred to as "cold disaster". Cold sea temperatures below the critical minimum for fish could lead to a high fish mortality (Hsieh et al. 2008). The death of wild fish was at least 73 t, and 80% of cage aquaculture fish were damaged (Chang et al. 2013). Chang et al. (2009) used satellite SST images around the Penghu Islands to show that the minimum SST (12.6 °C) in February 2008 was lower than the February climatological temperature (20 °C). The strong northeast monsoon in the winter of 2008, associated with La Niña, may drive the cold China Coastal Current to intrude more southward into the southern TS and can even suppress the northward warm Kuroshio Branch Current intruding the TS (Chen et al. 2010; Lee et al. 2014). Liao et al. (2013) suggested that the cold disaster in 2008 can be divided into three stages: First, the branch of the China Coastal Current moved cold water from the western strait to the central strait; then, a strong northeast wind intensified the southwest current; and finally, cold water gradually retreated to the north due to weakened wind.

Specifically, cold disaster in historic records happened not only in 2008 but also in 2000 (Lu et al., 2012) and in 2011 (Chang et al., 2013). The cold disaster in 2008 is the most serious event among other events occurring in the Penghu Islands. Thus, reducing the negative consequences of damage is a major concern. Although numerous studies on the single event of a cold disaster in 2008 have been performed using satellite-observed data and numerical models, there is no operational system for rapidly transmitting current information on potential sea threats to mariculturists or citizens at large. The main purposes of the

current study were to assess the predictability of exceptionally cold water, that might potentially trigger a cold disaster in the TS, and to present a feasible warning system with respect to marine hazards around the Penghu Islands.

## 2 Data and methods

The optimum interpolation daily SST dataset from the National Oceanic and Atmospheric Administration (NOAA) are
calculated from blended analyses, which is derived by combining multi-satellite data, ship observations, and buoy data (Reynolds et al. 2007). The dataset is averaged onto a $1/4° \times 1/4°$ spatial grid and covers the period from 1981 to the present; however, the current study used only the data from January 1995 to May 2017. In this study, we confined the analysis to the 60 coldest days of winter based on the climatologically averaged SST (January 1–March 1 in regular years, and January 1– February 29 in leap years; Supplementary Fig. 1). During these climatologically coldest days of winter (hereafter referred to
as just "winter days"), further cooling days may be expected to have the greatest implications for aquaculture.

The influence of temperature on fish is important not only in the surface layer but also in the subsurface layer. Although the methodology in the study is based on SST, it is excepted water column is well mixed in the vertical due to shallow bathymetry in the TS and the strong wind in winter. An insignificant temperature difference between surface and subsurface in this region ($< 1.2$ °C) is shown by the climatological temperature profile (Supplementary Fig. 2) during winter (averaged in December to
February 1985–2017) near the Penghu (23.75° N, 119.75° E). Thus, SST could be a suitable indicator depicting the water temperature of whole layers.

The Oceanic Niño Index (ONI) values, defined as a 3-month running mean of SST anomalies in the region of 5° N–5° S and 120°W–170°W, are taken from NOAA Climate Prediction Center (CPC; https://goo.gl/V6CtMD). The values are used for classifying the ENSO cycle into El Niño (ONI $\geq 0.5$ °C) and La Niña (ONI $\leq -0.5$ °C) (Huang et al. 2015). In addition, the
daily surface wind fields used in this study were derived from the National Centers for Environmental Prediction global analyses at 2.5° spatial resolution and can be applied from 1980 to the present, which was downloaded from https://www.esrl.noaa.gov/psd/data/gridded/data.ncep.html.

## 3 Cold events

The cold disaster is a biological or ecological response to low water temperature as defined in the introduction. The critical
temperatures inducing the death of different fishes are not consistent. Chang et al. (2013) indicated the activity of reef fish is declined at water temperature lower than 16 °C. Feeding activity of Cobia, a major species of cage aquaculture fish around Penghu Islands, is declined as lower than 18 °C and fish may die as lower than 15 °C (Lu et al., 2012). The sophisticated prediction for the cold disaster requires the understanding of the detailed physical and biological processes and will need the information about the marine resource. Unfortunately, we don't have data associated with a marine resource or aquaculture
production. The most relevant information is the date of occurrence of cold disasters in 2000, 2008 and 2011, indicating from the previous literature (Chang et al., 2013; Lu et al., 2012). Therefore, we won't focus on exploring the value of critical temperature. Instead, a goal of this paper is developing a warning system to predict the exceptionally cold water around the

Penghu Islands in the southern Taiwan Strait. It is expected that the presence of the exceptionally cold water points to the high possibility of the occurrence of a cold disaster (referring to the events of a large amount of fish death).

During the winter days (60 coldest days of winter as defined in Section 2), further cooling days are expected to happen the exceptionally cold water which may have the greatest implications for aquaculture (hereafter referred to "cold water days").

This study focused on hindcasting the occurrence of cold water days during the winter days. Regarding that the long-term observations of water temperature around Penghu are absent (~20-year time series needed), the cold water days were characterized by remotely sensed SST anomalies (SSTA) lower than a threshold (SSTA<-2.5 °C, i.e. SST< about 17 °C). SSTA is a deviation from the daily climatological average (Supplementary Fig. 1) and the threshold is estimated by 1.6 times the standard deviation (approximately 95% interval of the normal distribution) of SSTA. To obtain a quantity representative

of the magnitude of low SSTA, we selected the targeting area as a box in 23.5–24.5° N and 119–120° E (the white dash rectangle in Fig. 1), mainly off north coast of Penghu Island, covering the coolest SST deviation feature in Figure 3c, and a high correlation (r=0.94, p<0.05) with observational water temperature (Supplementary Fig. 3). Figure 2 shows the time series of SSTAs and highlights the SSTAs for the winter days (dots in Fig. 2a) from 1995 to 2017. Moreover, three or more consecutive cold water days were grouped into cold events; if any cold events were less than 4 days apart, they were grouped

into the same event. According to these definitions, 1,380 winter days of the study period, 95 cold water days and a total of 9 cold events (triangles in Fig. 2; Table 1) were observed. All of the cold events revealed in Figure 2 were determined to occur during the La Niña events. Furthermore, the cold phase peak of ENSO tends to occur toward the end of a year and a lag correlation reflects the cold events in January–February after the negative peak of ENSO. A similar lag correlation (0–6 months) between rainfall anomalies and ONI values during the La Niña events was observed in western Pacific (Wang et al., 2000).

Kuo et al. (2017) indicated SST in the TS was warming with a trend of about 0.15 °C/year during the period between 1980 and 2000. The possible interaction between the warming trend and cold events is unclear at this moment. However, the long-term trend of SST in the targeting area is gentle and its influence is insignificant (0.01 °C/year) during our studying period of 1995–2017.

To clarify the interannual variability of cold events in the southern TS, we performed composite analyses of SST and surface

wind fields for the winter days from 1995 to 2017. The spatial pattern of the long-term average for winter days was observed to show a moderate SST belt extending from southwest to northeast, and an isotherm of nearly 18 °C across the northern Penghu Islands was revealed to separate the colder water in the west from the warmer water in the southeastern TS (Fig. 3a). The cold-water China Coastal Current flows southwestward along the coast of China and meets the warm-water northward Kuroshio Branch Current near the Penghu Islands, forming a strong sea temperature front. The SST variability is obviously

affected by the delicate balance between the southward China Coastal Current and the northward Kuroshio Branch Current, both of which are associated with the magnitude of the northeast monsoon (Kuo and Ho 2004). Wind fields around Taiwan are strictly dominated by the East Asian monsoons. Wind data derived from weather stations across the TS (Jan et al. 2006) showed that a northeast monsoon occurs from September to the following May and that a southwest monsoon occurs during

the rest of the year. During the winter days, the northeast monsoon dominates the environmental conditions around Taiwan (Fig. 3d), which illustrates it can drive the cold SST front into the southern TS.

Figure 3b, c shows the SST deviation relative to the long-term average for the winter days. The SST across the TS was observed to get warmer along the China coast (Fig. 3b) when the northeast monsoon was weakened during the El Niño events, thus resulting in the wind anomaly fields illustrating southwest wind (Fig. 3h). By contrast, a negative SST deviation is shown to dominate all of the TS and to expand into the southern Penghu Islands during the La Niña events (Fig. 3c). The lowest deviation was approximately −0.6 °C near the central TS. In addition, a positive deviation regarding northeast wind was observed (Fig. 3i), which may intensify the northeast monsoon. The stronger wind generating turbulence mixing would likely lead to increased air-sea heat fluxes (Fig. 3f) and would substantially affect the extent of cold water in the TS. Figure 3d–3f shows surface total heat fluxes provided by NCEP/NCAR 40-year reanalysis project (Kalnay et al., 1996). More heat energy can be transferred from the ocean to the atmosphere in the La Niña (Fig. 3f) than that in the El Niño (Fig. 3e). These results imply that the SST variability in the TS is strongly associated with the ENSO cycle. Given the environmental conditions, exceptionally cold water is more likely to affect the southern TS and more potentially trigger cold disaster during La Niña than during El Niño events. According to above results, we summarize two possible physical mechanisms for triggering cold events in the TS. One is the balance between the southward China Coastal Current and the northward Kuroshio Branch Current, and the other one is local wind-driven entrainment. However, both processes are associated with the magnitude of wind. During the La Niña events (Fig. 4), a northeast monsoon dominates the environmental conditions around the TS with strong wind stresses. The cold China Coastal Current will have more chance to intrude into the southern TS and the warm Kuroshio Branch Current will even be suppressed by strong southwestward winds. In addition, strong wind stresses can drive turbulence mixing and enhance air-sea interaction to further cool the sea water across most of the TS. Therefore, ONI and local wind speed are used as prognostic indexes to find the cold days in the following study.

## 4 Predictability of cold events

### 4.1 Predicting by ONI

The results of the preceding analysis identify a significant correlation between cold water days and La Niña events. The association between them can be through the increased wind stress of the northeast monsoon, which may intensify the southwestward cold current. We extended this study by understanding the relationship with monthly ONI and evaluating the prediction skill as a function of lead time.

In this study, we primarily experiment with two-class prediction problems. The ENSO cycle was quantified using the ONI, and cold water days were predicted based on the ONI falling below a threshold. There are subsequently four possible results from the binary classification test: (1) the outcome predicted a cold water day is identical to the actual value (true positive, TP); however, (2) if the actual value is not a cold water day, it is classed as a false positive (FP). Conversely, (3) a true negative (TN) has been found while both the prediction state and the actual state are not cold water days; (4) the outcome predicted no

cold water day is exactly opposite to the actual value (false negative, FN). For the following evaluations, an ONI −0.9 was set as the threshold, and lead days were counted ahead of the cold day. For predictions at a 90-day lead time, ONI values below the threshold can be considered to correspond to the probability of cold water days occurring with a true positive rate (TPR, hit rate) of 72% and a false positive rate (FPR, false alarm rate) of 15% (the arrow in Fig. 5a). The hit rate is defined as

$\sum \text{TP}/[\sum \text{TP} + \sum \text{FN}]$ (i.e. the percentage of cold water days which are correctly identified as having the condition), and false

alarm rate is calculated as $\sum \text{FP}/[\sum \text{FP} + \sum \text{TN}]$. In addition, we could examine the suitability of a threshold at a special lead

time by an odds ratio, defined as $[\text{TPR}(1\text{-FPR})]/[\text{FPR}(1\text{-TPR})]$. In the above case, the odds ratio was 14.8, which implies the

probability of correct predictions is almost a 15-fold increase (the arrow in Fig. 5c).

Depending on the tolerance for the TPR and FPR, the choice of threshold for a prediction can be varied; more negative ONI

thresholds reduce both the TPR and FPR. This trade-off is presented by relative operating characteristic (ROC) curves (Hanley and McNeil 1982) that represent the relationship between the TPR and FPR as a function of threshold (Fig. 5a). ROC curves are quantified by integrating the area under the curve, called ROC score or area under the curve (AUC). The score is used as the proxy throughout the analysis because it is appropriate for assessing unusual events (Stephenson et al. 2008; McKinnon et al. 2016). When a ROC score is higher, it would be a more discriminating prediction method. In this study, the skill of the

ONI-based prediction was shown to peak at a lead time of 60 days, with a ROC score of 0.78, and generally decreased with increasing lead time. Moreover, significance can be estimated through the creation of a null distribution of the quantity of interest by using a block bootstrap (McKinnon et al. 2016), and significant results above the 95% confidence level are presented throughout this article. We considered a ROC score greater than 0.6 to be statistically significant in the predictions of cold water days. As shown in Fig. 5a, ROC scores decreased with lead time and were no longer significant by a lead time of 240

20    days. Notably, the ONI provided by NOAA CPC was estimated according to the 3-month running mean of monthly SSTAs in the Niño 3.4 region (5ºN–5ºS, 120º–170ºW). Because of the running mean needed, the ONI value has a delay time of two months; in other words, the latest ONI value obtainable in December is the value for October. Hence, the ONI-based prediction of cold water days can actually be employed with a lead time of 0–150 days. For simplicity, we still used the time of the ONI ahead of the cold water days to describe the lead time in the subsequent analysis.

Further, Saito and Rehmsmeier (2015) indicated the precision-recall curves (PRC) could be more informative than the ROC

curves on imbalanced datasets. PRC is a trade-off between positive predictive values (PPV), defined as $\sum \text{TP}/[\sum \text{TP} + \sum \text{FP}]$,

and TPR. Therefore, we also utilize the PRC to examine the predictability of cold water days (Fig. 5b). Although the baseline

of ROC is fixed, the baseline of PRC is decided by $[\sum \text{TP} + \sum \text{FN}]/[\sum \text{TP} + \sum \text{FN} + \sum \text{FP} + \sum \text{TN}]$. As a result, the baseline is

PPV=0.07 (a grey dashed line in Fig. 5b) and the prediction of cold water days become significant when the area under the

curve (AUC) of PRC is greater than 0.15. The results show that cold water days could be predicted with a lead time of 60–120 days, shorter than ROC method (60–210 days).

## 4.2 Predicting by wind

The relationship between wind stress and cold water can be understood through local wind-driven entrainment, whereby a strong La Niña episode results in an enhanced winter monsoon and strong wind stress increases turbulent mixing in favour of heat fluxes.

We next focused on the relationship between wind speed and the probability of cold water days. Wind speed variability across the TS could be quantified using an average wind speed (AWS) over a pre-specified period. The AWS was calculated using averaged periods of 1 to 30 days, and the correlation coefficients between AWS and SSTAs are shown in Fig. 6a. The 10-day averaged period was observed to have the highest correlation with the SSTA variability, and the correlation coefficient was approximately $-0.39$ ($p < 0.05$). Therefore, we focused on the AWS with a moving averaged period of 10 days in the subsequent analysis. Figure 6b shows the water temperature to drop following a stronger AWS.

As indicated by the ROC curves, the wind-based prediction had a ROC score of 0.8 at a lead time of 0 days, which is counted following the last day over which wind speed is averaged (Fig. 7). This result is consistent with the expected relationship between wind speed and sensible cooling. The wind-based predictions were observed to have significant prediction skill for lead times from 0 to 30 days (ROC score $\geq 0.6$), and the prediction skill generally decreased with increasing lead time. For predictions at a 15-day lead time, the highest odds ratio (6.0) was determined using a threshold of 11.5 m s$^{-1}$. An AWS below the special threshold was determined to correspond to the occurrence of cold water days with a TPR of 78% and an FPR of 38%. Moreover, the wind-based predictions were also examined through PRC (Fig. 7b). The results show that it has significant prediction skill for lead times from 0 to 25 days.

## 5 Warning mechanism for Penghu Islands

According to the analysis of predictability presented in the preceding section, cold water days can be predicted using the ONI for a long-lead prediction (60–120 days) and the AWS for middle- to short-lead predictions (0–25 days). We established a warning mechanism based on the ONI and AWS. The ONI-based prediction was employed to predict cold water days at a lead time of around 90 days. Hence, the ONI of $-0.9$, which engendered the highest accuracy (0.78) and odds ratio (14.8), was selected as the threshold. Moreover, the AWS-based prediction was employed for middle- and short-lead predictions. As shown in Fig. 8, the ROC scores varied with the integration periods and lead time. The prediction conducted at a lead time of around 15 days and an integration period of 10 days had a ROC score of approximately 0.74. The threshold was set at 11.5 m s$^{-1}$ because this was observed to result in the highest accuracy (0.70) and odds ratio (6.0). In addition, the prediction conducted at a lead time of around 5 days and integration periods of around 20 days had higher ROC scores compared with those conducted at other integration periods. Regarding the prediction of cold water days at an integration period of 20 days, the prediction conducted with a threshold of 12.5 m s$^{-1}$ had an accuracy of 0.76. Therefore, a warning mechanism could be

established based on this analysis (Table 2). The three warning thresholds mean various degree of risk with a different probability of occurrence.

**6 Hindcasting cold water days**

A hindcast of cold water days over the period 1995–2017 (Fig. 9) was obtained by using the warning mechanism in Table 2. The results clearly demonstrate high-risk warnings for the winters of 2000, 2008, 2011, and 2012. By monitoring the number of fish deaths around the Penghu Islands, Chang et al. (2013) and Lu et al. (2012) reported finding a large number of dead farmed fish in exceptionally cold water in the winters of 2000, 2008, and 2011. This agrees with the periods of high risk identified by this warning mechanism. Based on the results of hindcast and cold disasters in historic records (Chang et al., 2013; Lu et al., 2012), occurrence probabilities could be estimated within three warning thresholds (Table 2). For example, three of the high-risk years (red dots in Fig. 9) did indeed happen damage in historic records, indicating occurrence probability of damage is about 75% within a high-risk warning.

To illustrate specific predictions that could be made using the warning mechanism, we conducted a case study for the winter of 2011. Figure 10 displays the developing process of a cold water event from six satellite SST maps. Before the occurrence of the cold water event, the SST in the TS was 16 °C (Fig. 10a). An obvious cold front crossed from the Taiwan bank to the Chang-Yuen ridge and approached the Penghu Islands. A cold water (approximately 14 °C) developed along the coast of China on January 29, 2011 (Fig. 10b), and it extended to the southern TS afterward (Fig. 10c, d). When the exceptionally cold water intruded into the southern TS, the Penghu Islands were surrounded by extremely cold water below 16 °C. Finally, at the end of the cold event, warmer water above 22 °C was re-established through the channel to the east of Penghu Islands (Fig. 10e, f), and the cold water gradually retreated to the north due to the weakening wind (Fig. 10g). Furthermore, observational SST from buoy provided by the Central Weather Bureau of Taiwan (red star in Fig. 10a) was used to examine satellite SST. The temporal variation of satellite SST overall is similar to observational SST, but the magnitude of Satellite SST is often higher than that of observational SST (Fig. 10g). It should be noted that the coldest day detected by satellite SST have some time difference with the one detected by buoy SST. Even though the satellite SST used in this study is an optimum interpolation dataset combining multi-satellite data, the quality of SST is still low while over a long period of cloudy time. Because the most days of winter is under a cold front in TS often with a heavy coverage of clouds and the quality of satellite SST is sensitive to water vapour in the atmosphere, high-quality satellite SST is often not available in the TS during winter (Li et al., 2006). The result implies that observational water temperature is necessary when we would like to make a sophisticated prediction for the cold disaster. However, satellite SST in the targeting area could overall fit observational SST (r=0.94, p<0.05; Supplementary Fig. 3). It should be sufficient for the present goal for this work, to develop a warning system to predict the cold water days in the southern TS. Furthermore, the SST variability was observed to be associated with the 10-day AWS in Figure 10g. When the AWS remained at approximately 14 m s$^{-1}$ from January 10, the SST kept decreasing to approximately 14 °C until the AWS decreased on February 2. After wind speed weakening, the SST around the Penghu Islands rose immediately, which agrees

with the sequence of SST maps shown in Fig. 10d–f. A similar process was observed in the other cold events of 2000, 2008, and 2012.

SST measured by buoy provide a means for examining the applicability of the warning mechanism. The buoy (red star in Fig. 10a) measures SST after January 2007. The observational period includes three extremely cold winters whose warning level reached high risk: 2008, 2011, and 2012. Winter 2008 has the severest cold event over the past decade and the minimum SST (approximately 11 °C) broke the record temperature, discussed in numerous previous studies (e.g., Chen et al. 2010; Liao et al. 2013; Lee et al. 2014). Through the application of the warning mechanism, the hindcast of the 2008 winter showed that a high-risk warning could precede the coldest water day by 19 days (Fig. 11b); similarly, the high-risk warnings for 2000, 2011 and 2012 could lead by 10, 22 and 26 days, respectively. It should be noted that the warning lights in Fig. 11 are only shown from January to February because cold disasters in historical records are almost happening in February.

Figure 11 shows SST in 2008 and 2011 are lower than that in 2000 and 2012. The lowest SST appears in February in most years except in 2012. 10-day AWS stronger than 12 m s$^{-1}$ mainly appears from January to February in 2000, 2008 and 2011, but that appears from December to January in 2012; besides, AWS stronger than 14 m s$^{-1}$ maintains a longer time in 2008 and 2011. The results imply damage could be more serious in 2008 and 2011 (mentioned by Chang et al., 2013 and Lu et al., 2012) than in 2000 (mentioned by Lu et al., 2012). However, SST in winter of these four years all can be lower than 16 °C (Fig. 11), which is cold enough to induce the death of caged fish around Penghu Islands (Chang et al. 2013). Notably, the SST variability over a ~10-day period in February 2012 might be dominated by a sub-mesoscale process, which agrees with the higher correlation between the SST and 10-day AWS in Fig. 6a.

**7 Summary**

We used optimum interpolation SST data to identify exceptionally cold water over the southern TS during the period 1995–2017. The results reveal a total of 107 cold water days and 9 cold events likely to trigger cold disasters in the TS. Cold water develops along the coast of China and extends to the southern TS with sustained strong winds. In addition, these exceptionally cold water days always occur during La Niña events, and climatological maps show that the SST and wind speed over the TS are extremely low and strong in La Niña winters compared with normal or El Niño winters. A correlation was also obtained for SSTAs and 10-day AWS, with a correlation coefficient of $-0.39$ (p < 0.05).

According to the results associated with ENSO and wind speed, the predictability of cold water days can be estimated by ROC and PRC curves; when ROC scores were higher than or equal to 0.6, methodologies based on the ONI or 10-day AWS were observed to be significant (above the 95% confidence level) for predictions of cold water days. The ONI- and AWS-based predictions could be conducted at lead times of 60–120 and 0–25 days, respectively. Given this predictability, a possible warning mechanism based on the ONI and AWS was established. In this mechanism, if the monthly ONI is lower than or equal to $-0.9$, a cyan warning light indicating required action is triggered (~50% occurrence probability); this light turns yellow if the 10-day AWS is $\geq 11.5$ m s$^{-1}$, meaning a moderate risk of exceptionally cold water (~60% occurrence probability). Red light signaling high risk (~75% occurrence probability) displays when the 20-day AWS is $\geq 12.5$ m s$^{-1}$. After the application

of the warning mechanism, a hindcast of cold water days over the period 1995–2017 revealed four winters (2000, 2008, 2011, and 2012) with a high risk of a cold event potentially triggering damage. Three of the high-risk years (2000, 2008, and 2011) did indeed happen damage in historical records.

The warning mechanism evidently fulfills the requirements of the recently developed methodology of early warning systems for weather-related hazards, as implemented by the Central Weather Bureau of Taiwan. Warning lights based on the ONI and AWS indicate characteristics of exceptionally cold water affecting the Penghu Islands, and they thus facilitate identifying possible periods of exposure to extremely cold water. The warnings thus generated can be sent in an efficient and timely manner to mariculturists.

## Acknowledge

NCEP Daily Global Analyses data provided by the NOAA/OAR/ESRL PSD, Boulder, Colorado, USA, from their Web site at http://www.esrl.noaa.gov/psd/. NOAA High Resolution SST data provided by the NOAA/OAR/ESRL PSD, Boulder, Colorado, USA, from their Web site at https://www.esrl.noaa.gov/psd/. The Ocean Data Bank of the Ministry of Science and Technology, Taiwan, provided the historical CTD data and bathymetry data. The Central Weather Bureau (CWB) provided the buoy data sited near the Penghu Islands. Comments from two reviewers improved the preliminary manuscript markedly. This work was supported by the CWB of Taiwan through grant 1062076C and by the National Natural Science Foundation of China (U1405233).

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

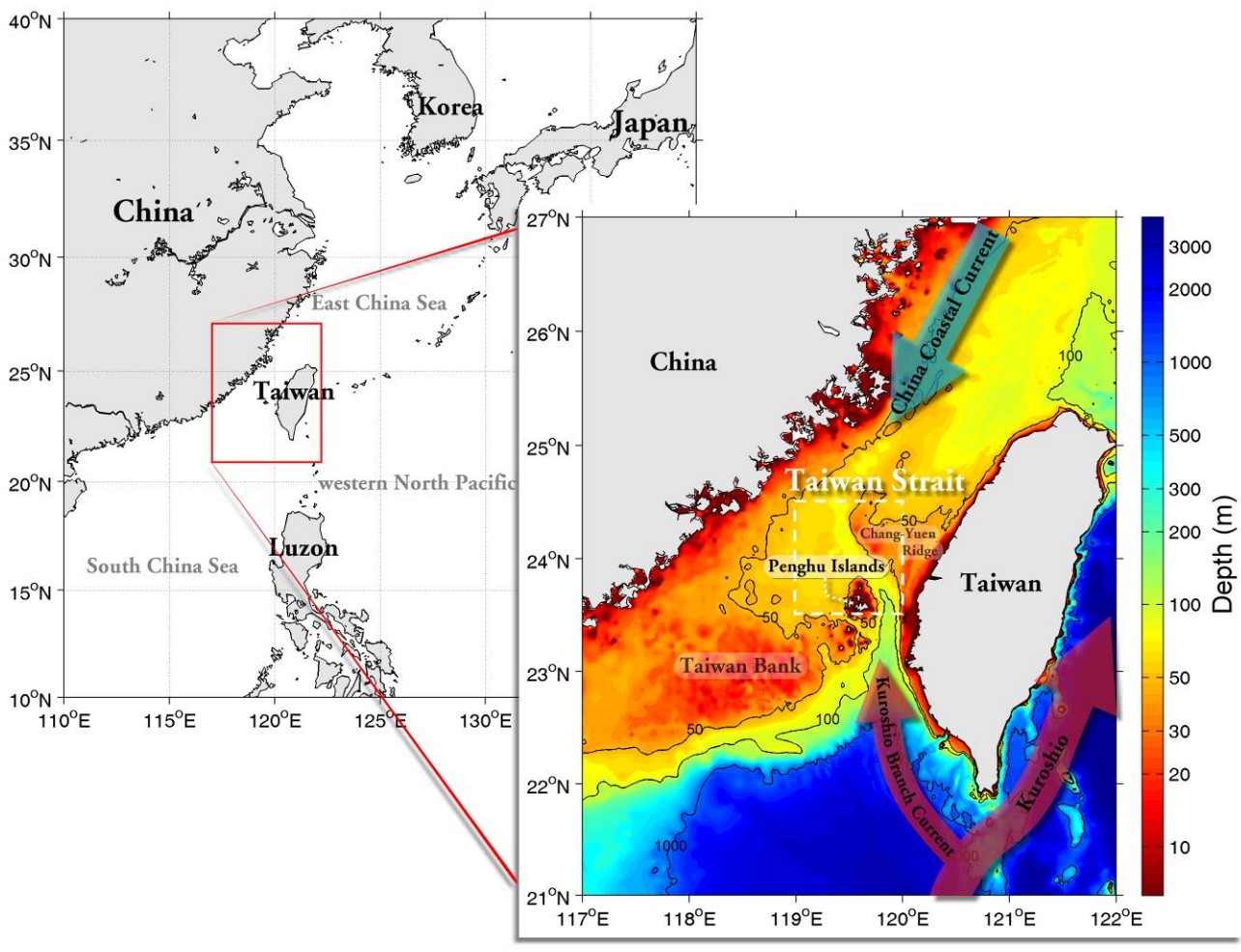

**Figure 1: Bathymetric chart (shaded color) and sketches of the China Coastal Current, Kuroshio, and Kuroshio Branch Current.**

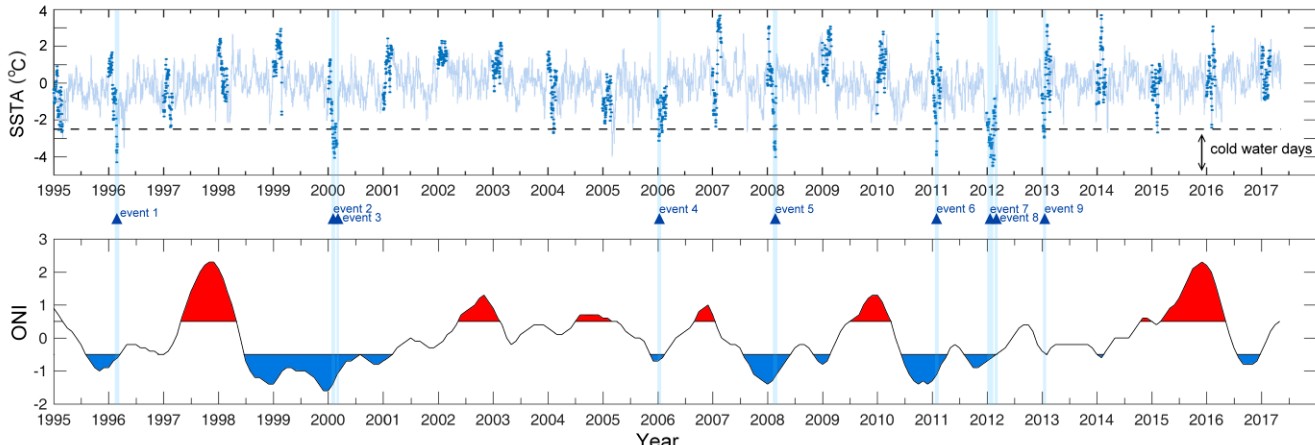

**Figure 2: (a) Time series of SSTAs (shading line) and that during the winter days are highlighted in dots. The dashed line denotes a magnitude of 1.6 times the standard deviation below the mean SSTA. (b) ONI time series within January 1995 and May 2017. Positive anomalies (≥0.5 °C, shaded red) indicate El Niño events, and negative anomalies (≤−0.5 °C, shaded blue) indicate La Niña events. Blue bars and triangles denote the occurrences of cold events.**

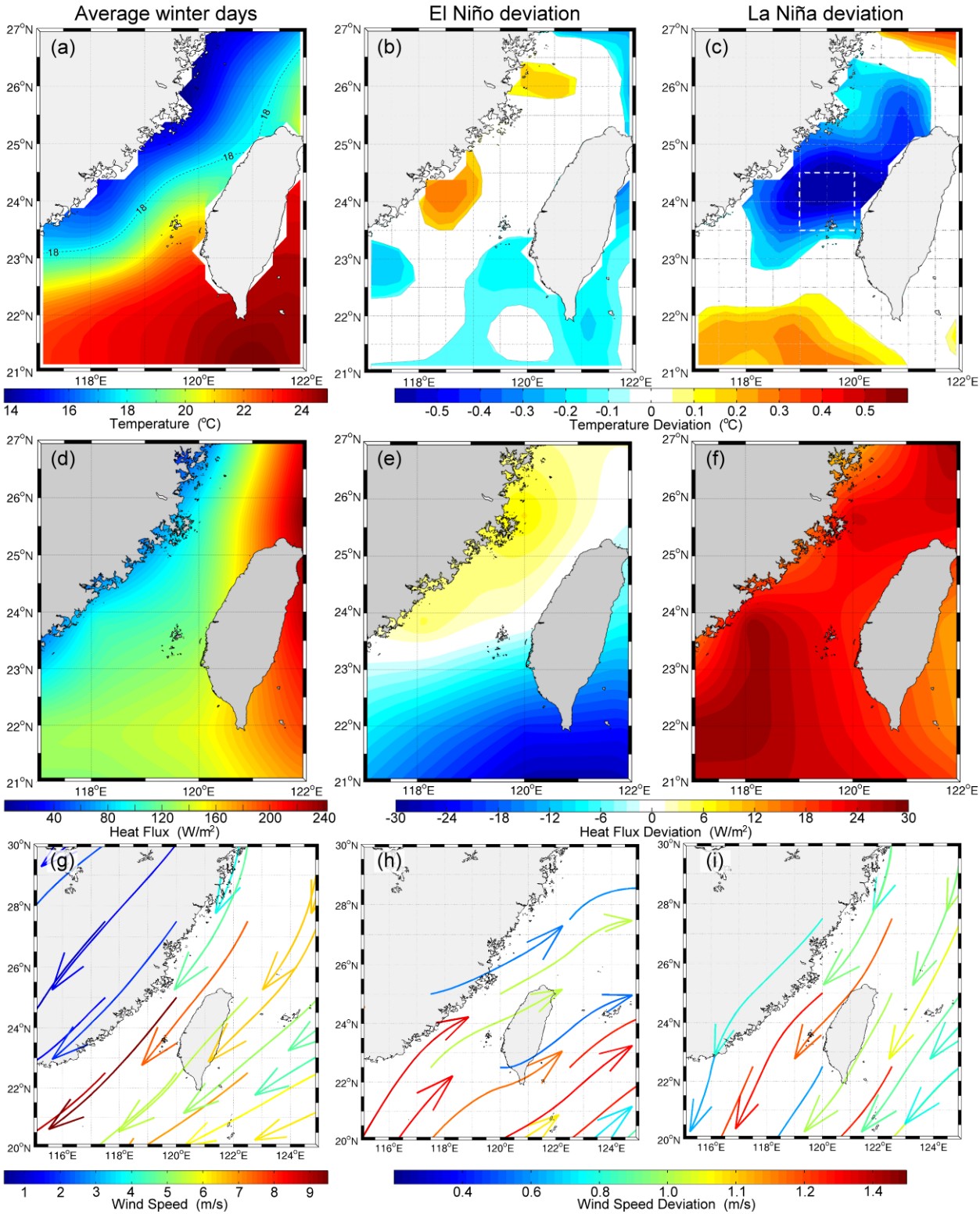

**Figure 3: Composite of SST (a to c), surface total heat fluxes (d to f), and surface wind fields (g to i) for (a, d, g) average winter days, (b, e, h) El Niño deviation, and (c, f, i) La Niña deviation. Only deviations above the 95% confidence level are shown. Positive heat flux values represent heat energy gain to the atmosphere.**

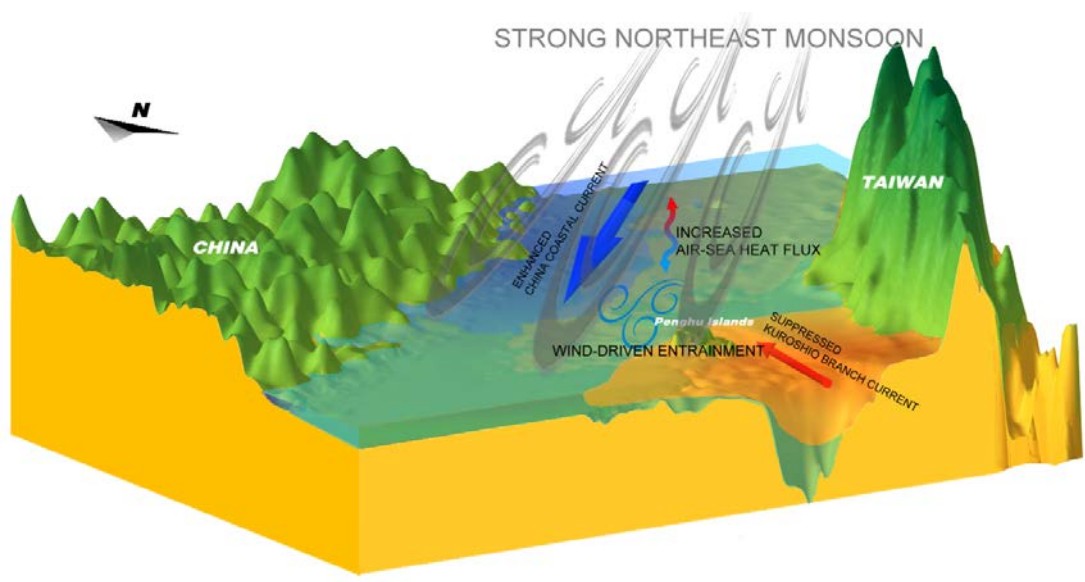

**Figure 4: A cold-event sketch during La Niña events.**

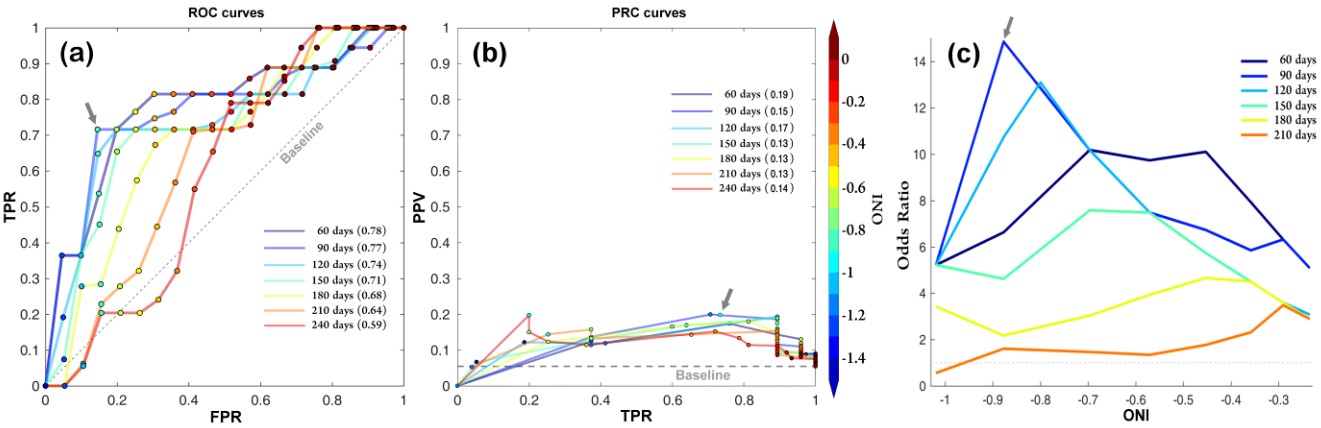

**Figure 5: (a) ROC curves for predicting cold water days at lead times of 60, 90, 120, 150, 180, 210, and 240 days using the negative**
10 **ONI. Numbers in parentheses in the legend are ROC scores for each lead time. The thresholds used to calculate TPRs and FPRs are showed from the 100th (lower left squares) to the 0th (upper right squares) percentile of negative ONI in decrements of 5%. ONIs at the dots are indexed by the colorbar. ROC scores ≥ 0.6 indicate a significant (p < 0.05) proxy for predictability. (b) As in Fig. 5(a) but plotting PRC curves. AUC of PRC ≥ 0.15 indicate a significant (p < 0.05) proxy for predictability. (c) Odds ratios vary with ONI thresholds at lead times of 60, 90, 120, 150, 180, 210, and 240 days. Arrows denote the point with a -0.9 threshold at a 90-day lead**
15 **time.**

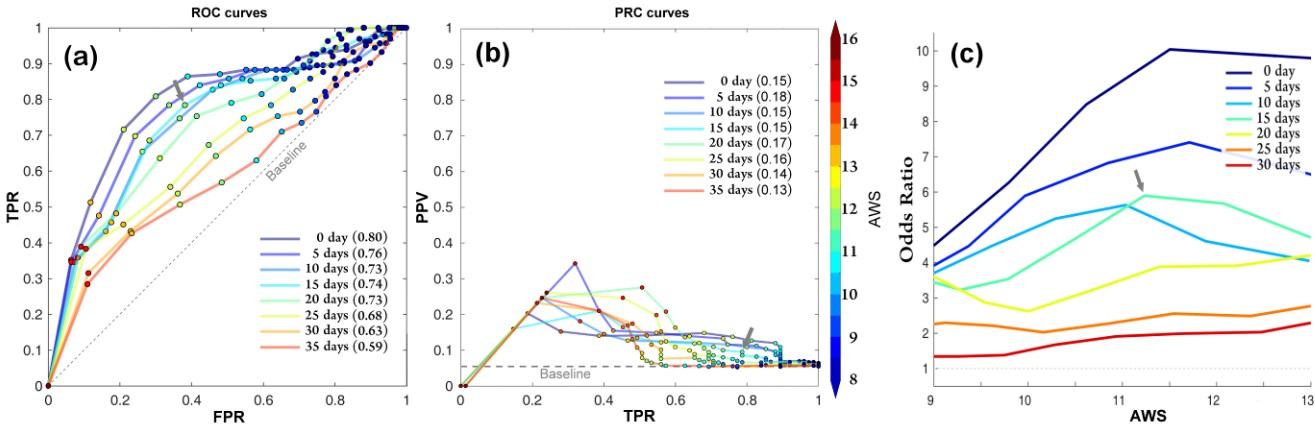

**Figure 6: (a) Correlation coefficients varied with integration time. (b) Time series during the winter days of SSTAs and AWS with an averaged time of 10 days.**

Figure 7 (a) ROC curves: 0 day (0.80), 5 days (0.76), 10 days (0.73), 15 days (0.74), 20 days (0.73), 25 days (0.68), 30 days (0.63), 35 days (0.59)

Figure 7 (b) PRC curves: 0 day (0.15), 5 days (0.18), 10 days (0.15), 15 days (0.15), 20 days (0.17), 25 days (0.16), 30 days (0.14), 35 days (0.13)

Figure 7 (c): 0 day, 5 days, 10 days, 15 days, 20 days, 25 days, 30 days

**Figure 7: As in Fig. 5, but for predictions of cold events at lead times of 0, 5, 10, 15, 20, 25, 30, and 35 days using the wind speed with an averaged time of 10 days. Arrows denote the point with an 11.5-m s⁻¹ threshold at a 15-day lead time.**

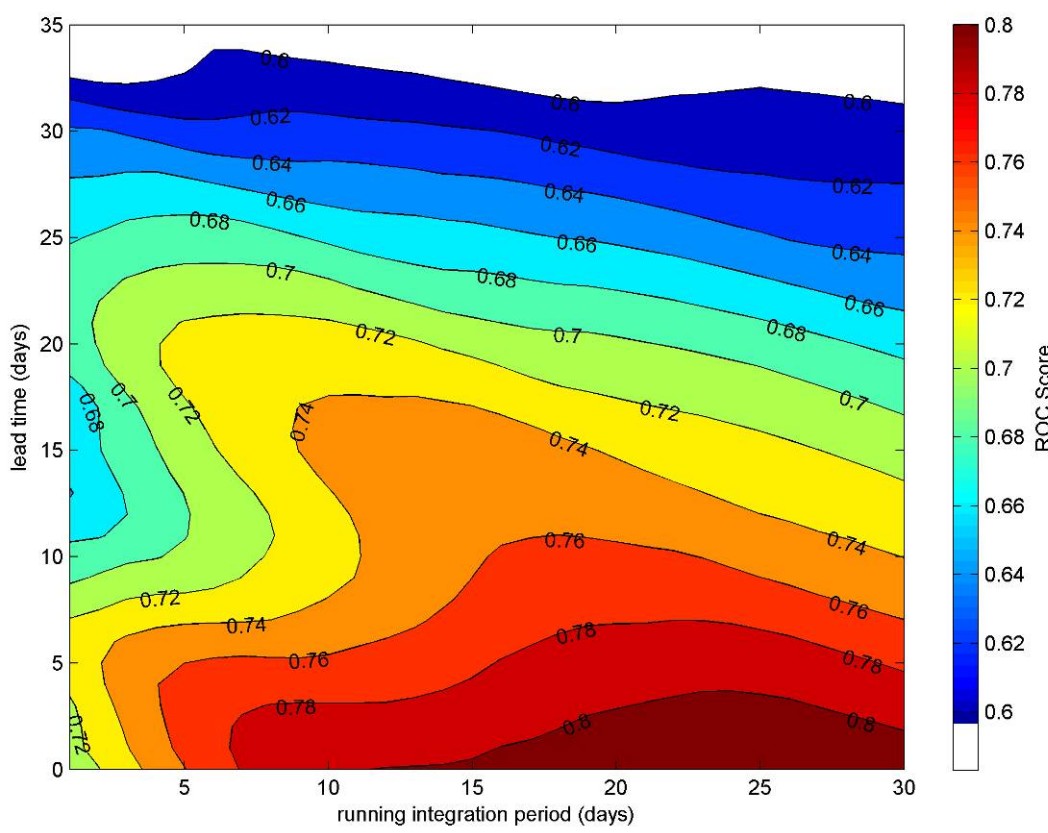

**Figure 8: ROC score diagram of AWS-based prediction. The white color masks nonsignificant regions, where ROC scores are below the 95% confidence level.**

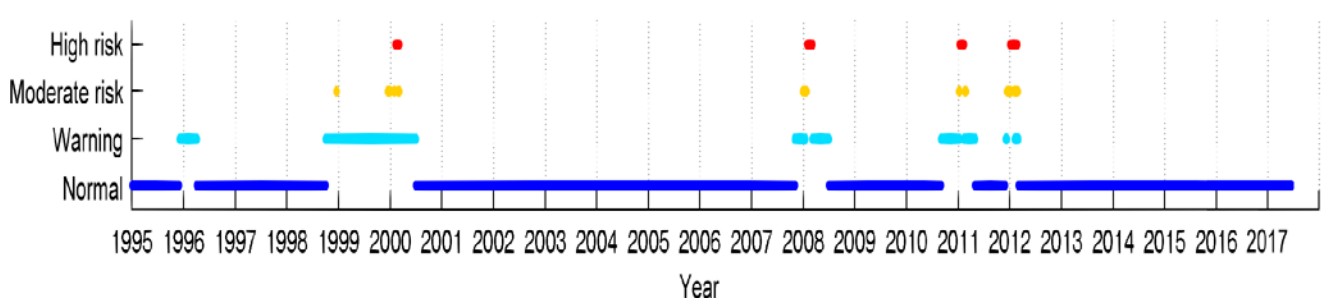

**Figure 9: Warning lights on cold water days for Penghu Islands.**

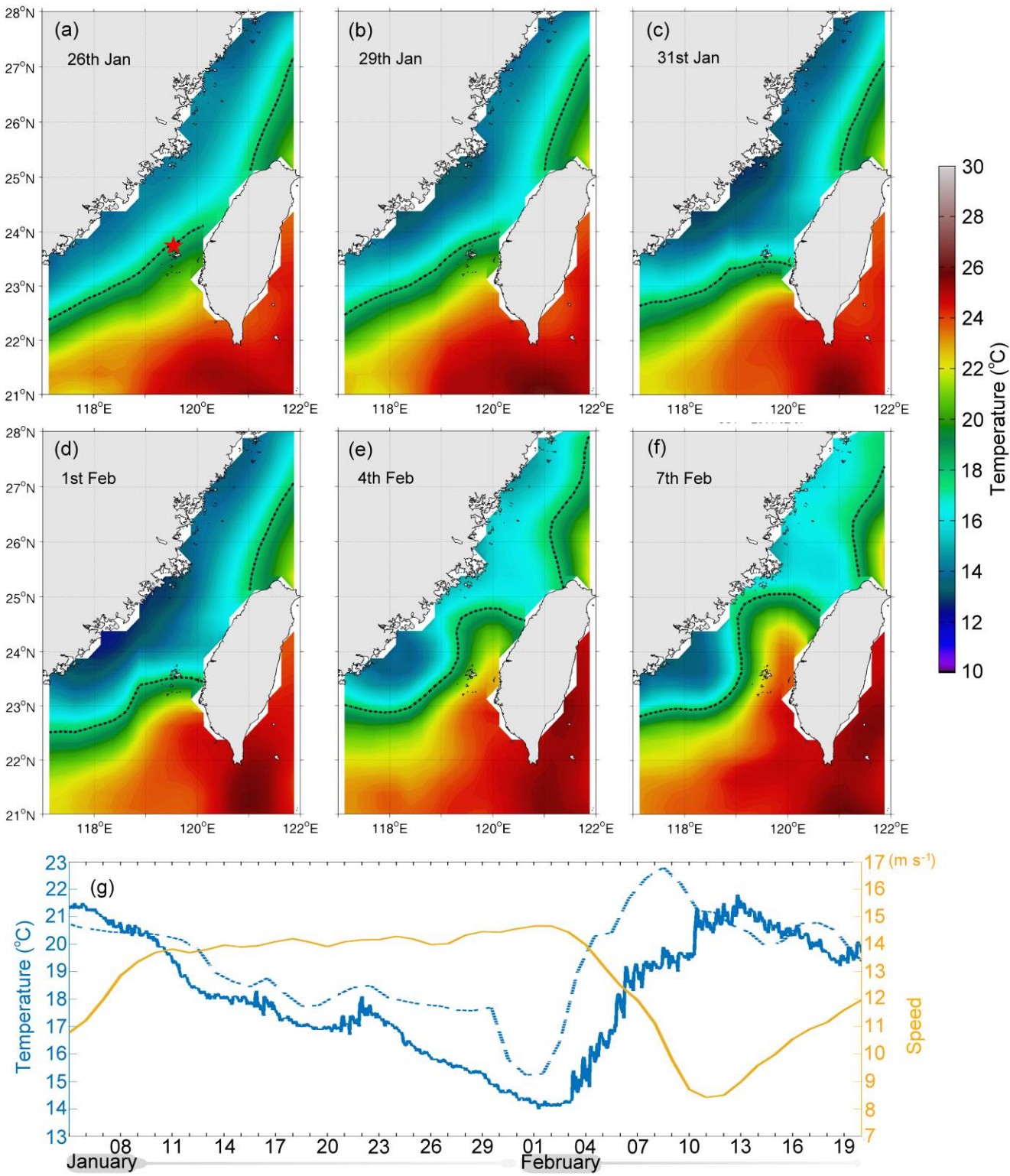

**Figure 10: A cold event in the winter of 2011. (a to f) A map sequence of satellite SST. The red star indicates the position of the buoy from the Central Weather Bureau of Taiwan. Dashed lines are isotherm lines of 18 °C. (g) Time series of satellite SST (blue dashed line), buoy SST (blue line), and 10-day AWS (yellow line) on the red star location from January 5 to February 20.**

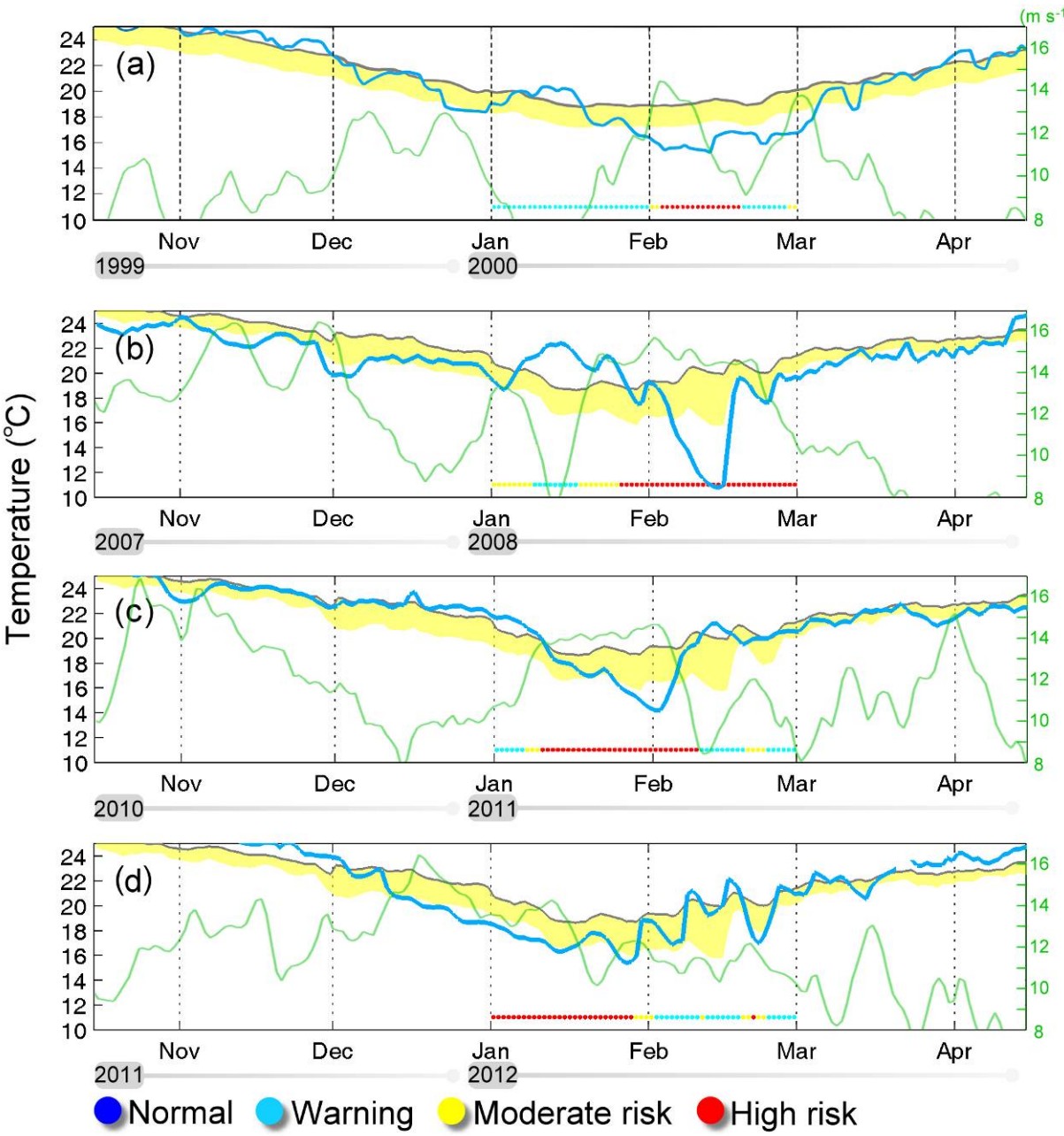

**Figure 11: Cold events in (a) 2000, (b) 2008, (c) 2011 and (d) 2012. Blue line: SST observed (a) by satellite and (b, c, d) by buoy; gray line: 10-year climatological average; yellow shading: range of standard deviation below the average; green line: 10-day AWS. Color dots are warning lights.**

**Table 1: A list of cold events during 1995-2017**

| Number | Date (yyyy/mm/dd) |
|--------|-------------------|
| Event 1 | 1996/02/23－1996/02/29 |
| Event 2 | 2000/02/02－2000/02/15 |
| Event 3 | 2000/02/23－2000/02/29 |
| Event 4 | 2006/01/09－2006/01/14 |
| Event 5 | 2008/02/16－2008/02/25 |
| Event 6 | 2011/01/30－2011/02/02 |
| Event 7 | 2012/01/16－2012/02/12 |
| Event 8 | 2012/02/18－2012/02/20 |
| Event 9 | 2013/01/16－2013/01/19 |

**Table 2: Warning thresholds suggested for exceptionally cold water days**

| Type | Conditions | Possible Occurrence Time | Probability of Occurrence |
|------|-----------|--------------------------|---------------------------|
| Warning | ONI $\leq$ -0.9 | around the next 90 days (30 days[a]) | 50% |
| Moderate risk | 10-day AWS $\geq$ 11.5 | around the next 15 days | 60% |
| High risk | 20-day AWS $\geq$ 12.5 | around the next 5 days | 75% |

5  [a] real lead time considering when the ONI value can be obtained.