# Peer review of "Exceptionally cold water days in the southern Taiwan Strait: their predictability and relation to La Niña"

_Natural Hazards and Earth System Sciences, 2017_

## Referee Comment (RC1) · Anonymous Referee #1 · 7 Feb 2018

This is an interesting research. The aim of this work is to develop a warming system for fishery to predict exceptionally cold water days in the southern Taiwan Strait. The authors used ONI and wind speed as indicators to predict the days and found that both proxies can be at lead times of 60 - 210 days and 0 - 30 days, respectively. This analysis results are useful for the regional warming system and worth publishing. Thus, this reviewer recommends the manuscript to be accepted for publishing after doing the following minor revisions.

1. The aim of this study is to assess the predictability of exceptionally cold water in the Taiwan Strait and to develop a warning system. Therefore, the tests conducted using

relative operating characteristics curves (ROC) need to be careful because ROC plots could be misleading when applied in imbalanced classification scenarios. However, Precision/Recall (PRC) plots can provide an accurate prediction of future classification performance (Saito and Rehmsmeier, 2015, The Precision-Recall Plot Is More Informative than the ROC Plot When Evaluating Binary Classifiers on Imbalanced Datasets, PLOS one). This reviewer suggests the authors apply PRC plots to confirm the predictability.

2. Page 2, lines 2-3. It is better to have some references to support the statement.

3. Page 2, lines 20-21. The critical temperatures for different fished are different. What is the critical temperature defined in this study for exceptionally cold water?

4. Page 2, line 18. Does the winter of 2008 mean from December 2007 to January 2008 or from December 2008 to January 2009? Please make it clear.

5. Page 3, line 4. This study used sea surface temperature (SST) as the indicator of temperature. However, the influence of temperature on fish is not only SST but also the temperature at subsurface layer. Is the temperature at sea surface and subsurface the same in the study area?

6. Page 3, line 25. Is it 1320 winter days or 1380? Please check and confirm it. If the span of data is from January 1995 to May 2007, the reviewer's calculation is 1380?

7. The authors gave the ONI time series in Figure 2. Are these ONI values calculated by the authors self or an official data from NOAA CPC? If it was calculated by authors, it's better to indicate the relative time period for calculating the SSTA?

8. Page 4, line 17. It better to show and discuss the results of air-sea heat fluxes in El Niño and La Niña events instead of just giving the temperature difference in Figure 3.

9. Is it possible to list all during dates of "cold events" in a new table? Figure 2 (a) does not show clearly, for example, events 2 and 3, and events 7 and 8.

[Figure]

10. The English of the manuscript is understandable, but still needs to be carefully polished.

---

## Referee Comment (RC2) · Anonymous Referee #2 · 9 Feb 2018

It is very delighted to see these authors to describing the Exceptionally cold water days in the southern Taiwan Strait: their predictability and relation to La Niña. This manuscript tried to assess the predictability of exceptionally cold water and to develop a warning system in the Taiwan Strait (TS). It was clearly written, and already to develop one warning system using the Oceanic Niño Index and integrated wind speed. But it is still difficult to apprehend whether the authors were mainly concerned of "Exceptionally cold water days in the southern Taiwan Strait". Firstly, the authors need to clearly define the cold waters days or the hotspot area (such as Penghu islands) as they demonstrate exceptionally cold water hit the marine natural resources around the Penghu Islands in the southwestern TS, causing considerable damage in marine

aquaculture. Secondly, "cold damage" is still unclear. Based on the description in this submission, the "cold damage" should be a kind of biological or ecological response to low water temperature in the waters around Penghu islands. Therefore, the authors need to consider where is the optimum area for developing the warning system on "Cold damage". And, the analysis or observation on the impact of marine resource or aquaculture production of hotspot area (NOT equal to the blue dotted quadrilateral in Figure 1) may important in the session of result or discussion. For example, the author showed a moderate SST belt extending from southwest to northeast, and an isotherm of nearly 18 °C across the northern Penghu Islands in fig.3a. It might reveal to separate the colder water in the west from the warmer water in the waters of northern Penghu Islands (Not in the southeastern TS). Thirdly, risk definition is also unclear. I did not know whether the risk include both of the vulnerability and impacts. I was also tried to search similar report for Coral Bleaching Products of NOAA (http://www.ospo.noaa.gov/Products/ocean/coral_bleaching.html) for high risk influenced by the vulnerability and impacts . They indicate the accumulation of thermal stress (i.e. Degree Heating Weeks, DHWs) that coral reefs have experienced over the past 12 weeks. At the same time, they also define the magnitude of impact levels as : the minor (<25% affected), moderate (26–50% affected), and severe (>50% affected) bleaching responses observed at the study sites. If possible, please try to explain the risk in this manuscript. Fifthly, the warning system found the high risk (or hazards) happened in the years of 2000, 2008, 2011 and 2012. But the reference only indicates the "cold damage" happened in 2008 and 2011 (Chang et al., 2013), How about the condition in 2000 and 2012. I suppose there are weak cold damages in years of 2000 and 2012. If so, the authors should consider to explain or discuss about what's the difference of marine environment or wind condition between cold damage (2008 and 2011) and weak or non-cold damage (2000 and 2012) years. And if possible, please add one sub-figure for cold events in 2000 to compare the annual variations in figure 11, as the authors describe the cold damage around the Penghu Islands has occurred three times: 2000, 2008, and 2011 (line 30-31, page 2). Otherwise, the Oceanic Niño

Index (ONI), defined as a 3-month running mean of SST anomalies, is describe in the line 10, page 3. However, the ONI indexes in January, February and March 2012 are -0.8, -0.6 and -0.5, respectively. It seems that the 2012 winter did not match up with the first definition of <-0.9°C. Why? And whether the author is considered to describe or discuss about the long-term variation or trend in Taiwan Strait as the topic is focus on "Exceptionally cold water days". If so, a new publish was suggested as your reference "Kuo et al., 2017 or 2018, Long-term observation on sea surface temperature variability in the Taiwan Strait during the northeast monsoon season, International Journal of Remote Sensing".

Specific: 1. There are too many errors of reference forms in the reference sections. For example, the format in line 17 and 25 of page 9 for references are different as following: Line 17: Kuo N-J, Ho C-R (2004). . .. Line 19: Lau N-C, Nath MJ (2006). . . Line 25: Lu Yi-Lin, Hsien I-L, Chung C-S, Lin, C-Y, Chen S-C, Tsai W-S (2012) Please check in detail by yourself. 2. L19∼20 of page 1, The authors may consider to modify the geographic term, for example, the average depth is 50 m, as they also use the description "approximately 30 m" for the Taiwan Bank. 3. Please try to explain the importance of this sentence "A lag-0- to lag-6-month correlation between rainfall anomalies in western Pacific and the peak La Niña was also observed by Wang et al. (2000)." (line 29-30, page). Did author try to say something using this sentence. 4. L8∼L9 of page 3, the authors use "the 60 coldest days of winter based on the climatologically averaged SST (January 6–March 6 in non-leap years, and January 6–March 5 in leap years)" is not easy to understand the coldest days. The authors may consider to add one figure or supplement figure for this. In addition, please confirm the definition of SSTA in line 17 of page 3. The sea surface temperature anomaly (SSTA) is the difference between the observed SST and the climatological SST. Did author use which the climatological daily SST is? In general, the SSTA is good indicator to see the long-term warming or cooling trend.
* * *
[Figure]

2017-414, 2018.

---

## Author Response (AR1)

**Response to Referee #1:**

Please find our responses below, with reviewer's comments in italics, authors' responses in standard format, and author's changes in manuscript in blue words.

**GENERAL COMMENT**

*This is an interesting research. The aim of this work is to develop a warming system for fishery to predict exceptionally cold water days in the southern Taiwan Strait. The authors used ONI and wind speed as indicators to predict the days and found that both proxies can be at lead times of 60 - 210 days and 0 - 30 days, respectively. This analysis results are useful for the regional warming system and worth publishing. Thus, this reviewer recommends the manuscript to be accepted for publishing after doing the following minor revisions.*

**Reply:**

We would like to thank the reviewer for providing constructive comments and suggestions.

1. *The aim of this study is to assess the predictability of exceptionally cold water in the Taiwan Strait and to develop a warning system. Therefore, the tests conducted using relative operating characteristics curves (ROC) need to be careful because ROC plots could be misleading when applied in imbalanced classification scenarios. However, Precision/Recall (PRC) plots can provide an accurate prediction of future classification performance (Saito and Rehmsmeier, 2015, The Precision-Recall Plot Is More Informative than the ROC Plot When Evaluating Binary Classifiers on Imbalanced Datasets, PLOS one). This reviewer suggests the authors apply PRC plots to confirm the predictability.*

**Reply:**

Thanks the reviewer for pointing this potential problem out. In a similar research work, McKinnon et al. (2016, Nature Geoscience) have used ROC curves to analyze SST and successfully predicted 321 hot days from the 2040 summer days in imbalanced classification scenarios. Both of ROC and PRC are statistical methods to find out a threshold depending on the tolerance for the TPR, FPR, and PPV. As suggested by the reviewer, we have incorporated PRC method in our revised manuscript. By using PRC method, the lead time are 60-120 days and 0-25 days for the ONI-based prediction and the wind-based predictions, respectively. We have added the consequent discussion in the revised manuscript.

Please see lines 23-29 in page 6 and lines 17-18 in page 7.

Further, Saito and Rehmsmeier (2015) indicated the precision-recall curves (PRC) could be more informative than the ROC curves on imbalanced datasets. PRC is a trade-off between positive predictive values (PPV), defined as $\sum_{TP}/[\sum_{TP}+\sum_{FP}]$, and TPR. Therefore, we also utilize the PRC to examine the predictability of cold water days (Fig. 5b). Although the baseline of ROC is fixed, the baseline of PRC is decided by $[\sum_{TP}+\sum_{FN}]/[\sum_{TP}+\sum_{FN}+\sum_{FP}+\sum_{TN}]$. As a result, the baseline is PPV=0.07 (a grey dashed line in Fig. 5b) and the prediction of cold water days become significant when the AUC of PRC is greater than 0.15. The results shows that cold water days could be predicted with a lead time of 60–120 days, shorter than ROC method (60–210 days).

[Figure]

Figure 5: (a) ROC curves for predicting cold water days at lead times of 60, 90, 120, 150, 180, 210, and 240 days using the negative ONI. Numbers in parentheses in the legend are ROC scores for each lead time. The thresholds used to calculate TPRs and FPRs are showed from the 100th (lower left squares) to the 0th (upper right squares) percentile of negative ONI in decrements of 5%. ONIs at the dots are indexed by the colorbar. ROC scores $\geq 0.6$ indicate a significant ($p < 0.05$) proxy for predictability. (b) As in Fig. 5(a) but plotting PRC curves. AUC of PRC $\geq 0.15$ indicate a significant ($p < 0.05$) proxy for predictability. (c) Odds ratios vary with ONI thresholds at lead times of 60, 90, 120, 150, 180, 210, and 240 days. Arrows denote the point with a -0.9 threshold at a 90-day lead time.

Moreover, the wind-based predictions were also examined through PRC (Fig. 7b). The results shows that it has significant prediction skill for lead times from 0 to 25 days.

[Figure]

Figure 7: As in Fig. 5, but for predictions of cold events at lead times of 0, 5, 10, 15, 20, 25, 30, and 35 days using the wind speed with an averaged time of 10 days. Arrows denote the point with an 11.5-m/s threshold at a 15-day lead time.

*2. Page 2, lines 2-3. It is better to have some references to support the statement.*

**Reply:**

Yes, two relevant papers, Wang et al. (2000) and Lau et al. (2006), were added in the revised manuscript. Please see lines 3-5 in page 2.

"It is known the cold phase of ENSO, La Niña, tends to intensify the East Asian winter monsoon, which often accompanies strong northerly winds and sharp air temperature drops. By contrast, the warm phase, El Niño, suppresses the East Asian winter monsoon (Wang et al., 2000; Lau et al., 2006)."

*3. Page 2, lines 20-21. The critical temperatures for different fished are different. What is the critical temperature defined in this study for exceptionally cold water?*

**Reply:**

Yes, we understand the critical temperatures inducing the death of different fishes are not consistent. We have incorporated the known critical temperature for the fish kind associated with the cold disaster event in the revision. However, for cold disaster prediction, the relationship between fish death and critical temperature is questionable because (1) no fish death occurring west of Penghu Island (onshore of mainland China), where the water temperature is much lower than near Penghu Island, and (2) the fish could escape from the cold water zone, where the water temperature reach critical value. There must be some unknown physical and biological processes and their interaction. As a result, we won't focus on exploring the value of critical temperature. Instead, this manuscript studies exceptionally cold water, as indicated in the title of this manuscript, which might potentially trigger "cold damage" (we will name it as cold disaster, referring to the events of large amount of fish death, in the revised ms) in the TS and assess the predictability of exceptionally cold water. In this

study, exceptionally cold water (cold water day) is defined by SSTAs < −2.5 °C, translating into temperature about 17°C. We have added a figure in the supplement. Please see lines 24-27 in page 3.

"The cold disaster is biological or ecological response to low water temperature as defined in introduction. The critical temperatures inducing the death of different fishes are not consistent. Chang et al. (2013) indicated the activity of reef fish is declined at water temperature lower than 16 °C. Feeding activity of Cobia, a major species of cage aquaculture fish around Penghu Islands, is declined as lower than 18 °C and fish may die as lower than 15 °C (Lu et al., 2012). The sophisticated prediction for the cold disaster require the understanding of the detailed physical and biological processes and will need the information about marine resource. Unfortunately, we don't have data associated with marine resource or aquaculture production. The most relevant information is the date of occurrence of cold disaster in 2000, 2008 and 2011, indicating from the previous literature (Chang et al., 2013; Lu et al., 2012). Therefore, we won't focus on exploring the value of critical temperature. Instead, a goal of this paper is developing a warning system to predict the exceptionally cold water around the Penghu Islands in the southern Taiwan Strait. It is expected that the presence of the exceptionally cold water points to the high possibility of the occurrence of cold disaster (referring to the events of large amount of fish death).

During the winter days (60 coldest days of winter as defined in Section 2), further cooling days are expected to happen the exceptionally cold water which may have the greatest implications for aquaculture (hereafter referred to "cold water days"). This study focused on hindcasting the occurrence of cold water days during the winter days. Regarding that the long-term observations of water temperature around Penghu are absent (~20–year time series needed), the cold water days were characterized by remotely sensed SST anomalies (SSTA) lower than a threshold (SSTA<-2.5 °C, i.e. SST< about 17°C). SSTA is a deviation from the daily climatological average (Supplementary Fig. 1) and the threshold is estimated by 1.6 times the standard deviation (approximately 95% interval of normal distribution) of SSTA. To obtain a quantity representative of the magnitude of low SSTA, we selected the targeting area as a box in 23.5–24.5°N and 119–120°E (the white dash rectangle in Fig. 1), mainly off north coast of Penghu Island, covering the coolest SST deviation feature in Figure 3c, and a high correlation (r=0.94, p<0.05) with observational water temperature (Supplementary Fig. 3)."

Figure S1

[Figure]

Figure S1: time series of climatological SST (black line) across the southern TS (the white dash rectangle in Figure 1) and the threshold of cold water days (gray line). Red dash lines reveal the period of winter days.

4. *Page 2, line 18. Does the winter of 2008 mean from December 2007 to January 2008 or from December 2008 to January 2009? Please make it clear.*

**Reply:**

The winter season in this area is from December to the following February. We have clarified it in the revised manuscript. Please see lines 18-20 in page 2.

"In the winter of 2008 (December 2007 to February 2008), exceptionally cold water affect the southern TS and hit the marine natural resources around the Penghu Islands in the southwestern TS, causing damage to the aquaculture and high fish mortality rates; this phenomenon is referred to as "cold disaster"."

5. *Page 3, line 4. This study used sea surface temperature (SST) as the indicator of temperature. However, the influence of temperature on fish is not only SST but also the temperature at subsurface layer. Is the temperature at sea surface and subsurface the same in the study area?*

**Reply:**

Thanks. Because the Taiwan Strait is shallow and the wind is very strong in winter, it is excepted water column is well mixed in the vertical. The climatological temperature profile during winter (averaged in December to February, 1985-2017) near the Penghu (23.75°N, 119.75°E) is displayed in Figure S2. Indeed, the figure showed insignificant temperature difference between surface and subsurface in this region (< 1.2°C). As a result, we believe SST is a suitable indicator depicting the water temperature of whole layers. We have added the above results in the revised manuscript. Please see lines 11-16 in page 3.

"The influence of temperature on fish is important not only in the surface layer but also in the subsurface layer. Although the methodology in the study is based on SST, it is excepted water column is well mixed in the vertical due to shallow bathymetry in the TS and the strong wind in winter. An insignificant temperature difference between surface and subsurface in this region (< 1.2 °C) is shown by the climatological temperature profile (Supplementary Fig. 2) during winter (averaged in December to February, 1985–2017) near the Penghu (23.75°N, 119.75°E). Thus, SST could be a suitable indicator depicting the water temperature of whole layers."

[Figure]

Figure S2: A climatological temperature profile during winter (December to February, 1985–2017) near the Penghu (23.75oN, 119.75oE), provided by the Ocean Data Bank of Taiwan.

6.   *Page 3, line 25. Is it 1320 winter days or 1380? Please check and confirm it. If the*

*span of data is from January 1995 to May 2007, the reviewer's calculation is 1380?*

**Reply:**

Thanks. It is 1380. We have corrected it in the revised manuscript. Please see lines 15-16 in page 4.

"According to these definitions, 1,380 winter days of the study period, 95 cold water days and a total of 9 cold events (triangles in Fig. 2; Table 1)"

7. *The authors gave the ONI time series in Figure 2. Are these ONI values calculated by the authors self or an official data from NOAA CPC? If it was calculated by authors, it's better to indicate the relative time period for calculating the SSTA?*

**Reply:**

ONI used in this manuscript are downloaded from NOAA CPC (https://goo.gl/V6CtMD). We have added essential illustration in the revised version. Please see lines 17-19 in page 3.

"The Oceanic Niño Index (ONI) values, defined as a 3-month running mean of SST anomalies in the region of 5°N–5°S and 120°W–170°W, are taken from NOAA Climate Prediction Center (CPC; https://goo.gl/V6CtMD). The values are used for classifying the ENSO cycle into El Niño (ONI ≥ 0.5 °C) and La Niña (ONI ≤ −0.5 °C) (Huang et al. 2015)."

8. *Page 4, line 17. It better to show and discuss the results of air-sea heat fluxes in El Niño and La Niña events instead of just giving the temperature difference in Figure 3.*

**Reply:**

Thanks for the reviewer's suggestions. We have added figures of heat flux and associated discussion in the revised manuscript. Please see lines 6-9 in page 5.

"The stronger wind generating turbulence mixing would likely lead to increased air–sea heat fluxes (Fig. 3f) and would substantially affect the extent of cold water in the TS. Figure 3d–3f shows surface total heat fluxes provided by NCEP/NCAR 40-year reanalysis project (Kalnay et al., 1996). More heat energy can be transferred from the ocean to the atmosphere in the La Niña (Fig. 3f) than that in the El Niño (Fig. 3e)."

[Figure]

Figure 3: Composite of SST (a to c), surface total heat fluxes (d to f), and surface wind fields (g to i) for (a, d, g) average winter days, (b, e, h) El Niño deviation, and (c, f, i) La Niña deviation. Only deviations above the 95% confidence level are shown. Positive heat flux values represent heat energy gain to the atmosphere.

9. *Is it possible to list all during dates of "cold events" in a new table? Figure 2 (a) does not show clearly, for example, events 2 and 3, and events 7 and 8.*

**Reply:**

Thanks for reviser's suggestion. We have clarified it in Table 1 in the revised manuscript.

**Table 1: A list of cold events during 1995-2017**

| Number | Date (yyyy/mm/dd) |
|--------|---------------------|
| Event 1 | 1996/02/23－1996/02/29 |
| Event 2 | 2000/02/02－2000/02/15 |
| Event 3 | 2000/02/23－2000/02/29 |
| Event 4 | 2006/01/09－2006/01/14 |
| Event 5 | 2008/02/11－2008/02/20 |
| Event 6 | 2011/01/30－2011/02/02 |
| Event 7 | 2012/01/16－2012/02/12 |
| Event 8 | 2012/02/18－2012/02/20 |
| Event 9 | 2013/01/16－2013/01/19 |

*10. The English of the manuscript is understandable, but still needs to be carefully polished.*

**Reply:**

Thanks for the suggestion. The manuscript has been through English editing before submission. We will revise it more carefully in the revision.

**Response to Referee #2:**

Please find our responses below, with reviewer's comments in italics, authors' responses in standard format, and author's changes in manuscript in blue words.

**GENERAL COMMENT**

*It is very delighted to see these authors to describing the Exceptionally cold water days in the southern Taiwan Strait: their predictability and relation to La Niña. This manuscript tried to assess the predictability of exceptionally cold water and to develop a warning system in the Taiwan Strait (TS). It was clearly written, and already to develop one warning system using the Oceanic Niño Index and integrated wind speed. But it is still difficult to apprehend whether the authors were mainly concerned of "Exceptionally cold water days in the southern Taiwan Strait".*

**Reply:**

We would like to thank the reviewer for providing constructive comments and suggestions.

1. *Firstly, the authors need to clearly define the cold waters days or the hotspot area (such as Penghu islands) as they demonstrate exceptionally cold water hit the marine natural resources around the Penghu Islands in the southwestern TS, causing considerable damage in marine aquaculture.*

**Reply:**

As mentioned in the Sections 2 and 3 of manuscript, cold water days in this manuscript are defined as SSTAs < −2.5 °C, i.e. the temperature is lower than about 17°C (we have added a figure to further explain in the Supplementary). SSTA is a deviation from the daily climatological average. In a similar research work, McKinnon et al. (2016, Nature Geoscience) have used the same method to analyze SST and successfully predicted extreme hot days in summer in US. The hot spot area has been re-defined as suggested by the reviewer. We will illustrate in the reply to the reviewer's second point.

Please see lines 5-9 page 4.

"Regarding that the long-term observations of water temperature around Penghu are absent (~20–year time series needed), the cold water days were characterized by remotely sensed SST anomalies (SSTA) lower than a threshold (SSTA<-2.5 °C, i.e. SST< about 17°C). SSTA is a deviation from the daily climatological average (Supplementary Fig. 1) and the threshold is estimated by 1.6 times the standard deviation (approximately 95% interval of normal distribution) of SSTA."

Figure S1

[Figure]

**Figure S1:** time series of climatological SST (black line) across the southern TS (the white dash rectangle in Figure 1) and critical temperature of cold water days (gray line). Red dash lines reveal the period of winter days.

2. *Secondly, "cold damage" is still unclear. Based on the description in this submission, the "cold damage" should be a kind of biological or ecological response to low water temperature in the waters around Penghu islands. Therefore, the authors need to consider where is the optimum area for developing the warning system on "Cold damage". And, the analysis or observation on the impact of marine resource or aquaculture production of hotspot area (NOT equal to the blue dotted quadrilateral in Figure 1) may important in the session of result or discussion. For example, the author showed a moderate SST belt extending from southwest to northeast, and an isotherm of nearly 18 ∘C across the northern Penghu Islands in fig.3a. It might reveal to separate the colder water in the west from the warmer water in the waters of northern Penghu Islands (Not in the southeastern TS).*

**Reply:**

Yes, the cold damage is biological or ecological response to low water temperature. To be specific, in the revised ms, we have defined "cold disaster" referring to the serious

fish death induced by exceptionally cold water around the Penghu Island. In this ms, we aim to develop a warning system to predict the cold water days in the southern Taiwan Strait, as indicated in the title of our manuscript. It is expected that the presence of the cold water days points to the high possibility of the occurrence of cold disaster. Regarding that the long-term observations of water temperature around Penghu are absent (~20- year time series needed), the cold water days were characterized by remotely sensed SSTA lower than a threshold.

We agree with the reviewer that it is important to find the "optimum area" to calculate SSTA and evaluate its impact on biological environment for the development of the warning system. Unfortunately, we don't have data associated with marine resource or aquaculture production. The most relevant information is the date of occurrence of cold disaster in 2000, 2008 and 2011, indicating from the previous literature. The information should be sufficient for the present goal for this work, to predict the cold water days. But the sophisticated prediction for the cold disaster require the understanding of the detailed physical and biological processes and will need the information about marine resource. This is certainly our next goal.

As mentioned by the reviewer, the targeting area we selected covers a frontal area as shown in Figure 3a, which may not be suitable for the index of cold water days. We have re-selected the targeting area as a box in 23.5-24.5°N and 119-120°E, mainly off north coast of Penghu Island, covering the coolest SSTA feature in Figure 3c, and a high correlation (r=0.94, p<0.05) with observational water temperature (Figure S3). SST from buoy sited on the north of Penghu Islands (red star in Figure 10a) is the most suitable indicator monitoring the water temperature around Penghu Island, but unfortunately buoy SST can be used after January 2007 and lost efficacy in 2013-2016. As mentioned in the above reply, the long-term observations of water temperature around Penghu are absent. Although the satellite SST in the targeting area is overall higher than SST measured by the buoy (Figure S3), it has a high correlation with observational SST and should be sufficient for the present goal for this work.
Please see from line 24 in page 3 to line 12 in page 4.

"The cold disaster is biological or ecological response to low water temperature as defined in introduction. The critical temperatures inducing the death of different fishes are not consistent. Chang et al. (2013) indicated the activity of reef fish is declined at water temperature lower than 16 °C. Feeding activity of Cobia, a major species of cage aquaculture fish around Penghu Islands, is declined as lower than 18 °C and fish may die as lower than 15 °C (Lu et al., 2012). The sophisticated prediction for the cold disaster require the understanding of the detailed physical and biological processes

and will need the information about marine resource. Unfortunately, we don't have data associated with marine resource or aquaculture production. The most relevant information is the date of occurrence of cold disaster in 2000, 2008 and 2011, indicating from the previous literature (Chang et al., 2013; Lu et al., 2012). Therefore, we won't focus on exploring the value of critical temperature. Instead, a goal of this paper is developing a warning system to predict the exceptionally cold water around the Penghu Islands in the southern Taiwan Strait. It is expected that the presence of the exceptionally cold water points to the high possibility of the occurrence of cold disaster (referring to the events of large amount of fish death).

During the winter days (60 coldest days of winter as defined in Section 2), further cooling days are expected to happen the exceptionally cold water which may have the greatest implications for aquaculture (hereafter referred to "cold water days"). This study focused on hindcasting the occurrence of cold water days during the winter days. Regarding that the long-term observations of water temperature around Penghu are absent (~20–year time series needed), the cold water days were characterized by remotely sensed SST anomalies (SSTA) lower than a threshold (SSTA<-2.5 °C, i.e. SST< about 17°C). SSTA is a deviation from the daily climatological average (Supplementary Fig. 1) and the threshold is estimated by 1.6 times the standard deviation (approximately 95% interval of normal distribution) of SSTA. To obtain a quantity representative of the magnitude of low SSTA, we selected the targeting area as a box in 23.5–24.5°N and 119–120°E (the white dash rectangle in Fig. 1), mainly off north coast of Penghu Island, covering the coolest SST deviation feature in Figure 3c, and a high correlation (r=0.94, p<0.05) with observational water temperature (Supplementary Fig. 3)."

[Figure]

Figure 1: Bathymetric chart (shaded color) and sketches of the China Coastal Current, Kuroshio, and Kuroshio Branch Current.

[Figure]

Figure 3: Composite of SST (a to c), surface total heat fluxes (d to f), and surface wind fields (g to i) for (a, d, g) average winter days, (b, e, h) El Niño deviation, and (c, f, i) La Niña deviation. Only deviations above the 95% confidence level are shown. Positive heat flux values represent heat energy gain to the atmosphere.

[Figure]

Figure S3: Time series of SST observed by satellite (yellow line) and buoy (blue line). Please note that buoy SST can be used after January 2007 and therefore the figure shows the correlation between satellite SST and buoy SST after January 2007.

3. *Thirdly, risk definition is also unclear. I did not know whether the risk include both of the vulnerability and impacts. I was also tried to search similar report for Coral Bleaching Products of NOAA (http://www.ospo.noaa.gov/Products/ocean/coral_bleaching.html) for high risk influenced by the vulnerability and impacts. They indicate the accumulation of thermal stress (i.e. Degree Heating Weeks, DHWs) that coral reefs have experienced over the past 12 weeks. At the same time, they also define the magnitude of impact levels as : the minor (<25% affected), moderate (26–50% affected), and severe (>50% affected) bleaching responses observed at the study sites. If possible, please try to explain the risk in this manuscript.*

**Reply:**

Thanks for the suggestion. We can't define the magnitude of impact levels or impact area as done by NOAA because it requires large amount of biological and fishery data in the vast ocean. Three different risks in the manuscript mean three different probability of occurrence. The revised manuscript has estimated the occurrence probability within various degree of risk.

Please see lines 2-9 in page 3.

"A hindcast of cold water days over the period 1995–2017 (Fig. 9) was obtained by using the warning mechanism in Table 2. The results clearly demonstrate high-risk warnings for the winters of 2000, 2008, 2011, and 2012. By monitoring the number of fish deaths around the Penghu Islands, Chang et al. (2013) and Lu et al. (2012) reported finding a large number of dead farmed fish in exceptionally cold water in the winters of 2000, 2008, and 2011. This agrees with the periods of high risk identified by this warning mechanism. Based on the results of hindcast and cold disasters in historic records (Chang et al., 2013; Lu et al., 2012), occurrence probabilities could be estimated within three warning thresholds (Table 2). For example, three of the high-risk years (red dots in Fig. 9) did indeed happen damage in historic records, indicating

occurrence probability of damage is about 75% within a high-risk warning."

Table 2: Warning thresholds suggested for exceptionally cold water days

| Type | Conditions | Possible Occurrence Time | Probability of Occurrence |
|---|---|---|---|
| Warning | ONI ≤ -0.9 | around the next 90 days (30 days[a]) | 50% |
| Moderate risk | 10-day AWS ≥ 11.5 | around the next 15 days | 60% |
| High risk | 20-day AWS ≥ 12.5 | around the next 5 days | 75% |

[a] real lead time considering when the ONI value can be obtained.

4. *Fifthly, the warning system found the high risk (or hazards) happened in the years of 2000, 2008, 2011 and 2012. But the reference only indicates the "cold damage" happened in 2008 and 2011 (Chang et al., 2013), How about the condition in 2000 and 2012. I suppose there are weak cold damages in years of 2000 and 2012. If so, the authors should consider to explain or discuss about what's the difference of marine environment or wind condition between cold damage (2008 and 2011) and weak or non-cold damage (2000 and 2012) years. And if possible, please add one sub-figure for cold events in 2000 to compare the annual variations in figure 11, as the authors describe the cold damage around the Penghu Islands has occurred three times: 2000, 2008, and 2011 (line 30-31, page 2).*

**Reply:**

As mentioned in Section1 & 6, cold disaster in historic records happened not only in 2008 and 2011 (Chang et al., 2013; Lu et al., 2012) but also in 2000 (Lu et al., 2012). The manuscript studies exceptionally cold water, which might potentially trigger disaster in the TS. A hindcast by the warning system showed high-risk warnings for the winters of 2000, 2008, 2011, and 2012, but it doesn't necessarily mean cold disaster must happen in these years. The results indicated that cold disaster likely happen in these four years. Actually, three of the high-risk years did indeed happen damage (2000, 2008, and 2011; Chang et al., 2013, Lu et al., 2012) in historic records, indicating occurrence probability of damage is about 75% within a high-risk warning.

Because the SST shown in Fig. 11 is observed by a buoy working after 2007, we don't have SST data in 2000 (Figure S3). However, we have added a sub-figure of SST observed by satellite in 2000 and do some discussions.

Please see line 2-9 in page 3.

"Figure 11 shows SST in 2008 and 2011 are more lower than that in 2000 and 2012. The lowest SST appears in February in most years except in 2012. 10-day AWS stronger than 12 m/s mainly appears from January to February in 2000, 2008 and 2011, but that appears from December to January in 2012; besides, AWS stronger than 14 m/s

maintains a longer time in 2008 and 2011. The results imply damage could be more serious in 2008 and 2011 (mentioned by Chang et al., 2013 and Lu et al., 2012) than in 2000 (mentioned by Lu et al., 2012). However, SST in winter of these four years all can be lower than 16 °C (Fig. 11), which is cold enough to induced the death of caged fish around Penghu Islands (Chang et al. 2013). Notably, the SST variability over a ~10-day period in February 2012 might be dominated by a sub-mesoscale process, which agrees with the higher correlation between the SST and 10-day AWS in Fig. 6a."

[Figure]

Figure 11: Cold events in (a) 2000, (b) 2008, (c) 2011 and (d) 2012. Blue line: SST observed (a) by satellite and (b, c, d) by buoy; gray line: 10-year climatological average; yellow shading: range of standard deviation below the average; green line: 10-day AWS. Color dots are warning lights (only shown from January to February).

5. *Otherwise, the Oceanic Niño Index (ONI), defined as a 3-month running mean of SST anomalies, is describe in the line 10, page 3. However, the ONI indexes in January, February and March 2012 are -0.8, -0.6 and -0.5, respectively. It seems that the 2012 winter did not match up with the first definition of <-0.9 C. Why? And whether the author is considered to describe or discuss about the long-term variation or trend in Taiwan Strait as the topic is focus on Exceptionally cold water days". If so, a new publish was suggested as your reference "Kuo et al., 2017 or 2018, Long-term observation on sea surface temperature variability in the Taiwan Strait during the northeast monsoon season, International Journal of Remote Sensing".*

**Reply:**

Yes. As mentioned in Section 4.1 of the manuscript, ONI values used in this manuscript are downloaded from NOAA CPC. They are estimated according to the 3-month running mean of monthly SSTAs in the Nino3.4 region (https://goo.gl/XRFVM3). Because of the running mean needed, the ONI value has a delay time of two month; in other words, the latest ONI value obtainable in this month (April) is the value for February (as the AC2-Figure 2 screenshot shown). Actually, the ONI indexes used in January, February and March 2012 are -1.1(Nov.), -1.0(Dec.) and -0.8(Jan.), respectively.

| Year | DJF | JFM | FMA | MAM | AMJ | MJJ | JJA | JAS | ASO | SON | OND | NDJ |
|------|-----|-----|-----|-----|-----|-----|-----|-----|-----|-----|-----|-----|
| 2010 | 1.5 | 1.3 | 0.9 | 0.4 | -0.1 | -0.6 | -1.0 | -1.4 | -1.6 | -1.7 | -1.7 | -1.6 |
| 2011 | -1.4 | -1.1 | -0.8 | -0.6 | -0.5 | -0.4 | -0.5 | -0.7 | -0.9 | -1.1 | -1.1 | -1.0 |
| 2012 | -0.8 | -0.6 | -0.5 | -0.4 | -0.2 | 0.1 | 0.3 | 0.3 | 0.3 | 0.2 | 0.0 | -0.2 |
| 2013 | -0.4 | -0.3 | -0.2 | -0.2 | -0.3 | -0.3 | -0.4 | -0.4 | -0.3 | -0.2 | -0.2 | -0.3 |
| 2014 | -0.4 | -0.4 | -0.2 | 0.1 | 0.3 | 0.2 | 0.1 | 0.0 | 0.2 | 0.4 | 0.6 | 0.7 |
| 2015 | 0.6 | 0.6 | 0.6 | 0.8 | 1.0 | 1.2 | 1.5 | 1.8 | 2.1 | 2.4 | 2.5 | 2.6 |
| 2016 | 2.5 | 2.2 | 1.7 | 1.0 | 0.5 | 0.0 | -0.3 | -0.6 | -0.7 | -0.7 | -0.7 | -0.6 |
| 2017 | -0.3 | -0.1 | 0.1 | 0.3 | 0.4 | 0.4 | 0.2 | -0.1 | -0.4 | -0.7 | -0.9 | -1.0 |
| 2018 | -0.9 | -0.8 | | | | | | | | | | |

AC2-Figure 1. ONI values from https://goo.gl/XRFVM3

The trend in our studying region has a gentle slope (0.01°C/year), which is not significant, during the studying period of 1995-2017 (dark blue line in AC2-Figure 2). We have added the above results and Kuo et al. (2017) as reference.

[Figure]

AC2-Figure 2. time series of SSTA. Blue line is a trend from 1995 to 2017 (0.01°C/year); red line is a trend from 1995 to 2000 (0.21°C/year).

Please see lines 19-22 in page 4.

"Kuo et al. (2017) indicated SST in the TS was warming with a trend of about 0.15 °C/year during the period between 1980 and 2000. The possible interaction between the warming trend and cold events is unclear at this moment. However, the long-term trend of SST in the targeting area is gentle and its influence is insignificant (0.01 °C/year) during our studying period of 1995–2017."

**SPECIFIC COMMENT**

1. *There are too many errors of reference forms in the reference sections. For example, the format in line 17 and 25 of page 9 for references are different as following: Line 17: Kuo N-J, Ho C-R (2004). . .. Line 19: Lau N-C, Nath MJ (2006). . . Line 25: Lu Yi-Lin, Hsien I-L, Chung C-S, Lin, C-Y, Chen S-C, Tsai W-S (2012) Please check in detail by yourself.*

**Reply:**

Thanks. We have modified that in the revised manuscript.

2. *L19~20 of page 1, The authors may consider to modify the geographic term, for example, the average depth is 50 m, as they also use the description "approximately 30 m" for the Taiwan Bank.*

**Reply:**

"The average depth is 50 m" give a description of the Taiwan Strait rather than of the Taiwan Bank. We have clarified it in the revised manuscript.

Please see lines 21-22 in page 1.

"The average depth of TS is 50 m and two major shallow water regions, Taiwan Bank and Chang-Yuen Ridge, are about 30 m (Fig. 1)"

3. *Please try to explain the importance of this sentence "A lag-0- to lag-6-month correlation between rainfall anomalies in western Pacific and the peak La Niña was also observed by Wang et al. (2000)." (line 29-30, page). Did author try to say*

*something using this sentence.*

**Reply:**

We would like to mention a lag correlation is shown not only between cold event and La Niña but also between rainfall and La Niña. We have clarified it in the revised manuscript.

Please see lines 16-19 in page 4.

"All of the cold events revealed in Figure 2 were determined to occur during the La Niña events. Furthermore, the cold phase peak of ENSO tends to occur toward the end of a year and a lag correlation reflects the cold events in January–February after the negative peak of ENSO. A similar lag correlation (0–6 months) between rainfall anomalies and ONI values during the La Niña events was observed in western Pacific (Wang et al., 2000)."

4. *L8~L9 of page 3, the authors use "the 60 coldest days of winter based on the climatologically averaged SST (January 6–March 6 in non-leap years, and January 6–March 5 in leap years)" is not easy to understand the coldest days. The authors may consider to add one figure or supplement figure for this.*

**Reply:**

Thanks. We have added a figure in the Supplementary. Please note that the 60 coldest days of winter has been modified as January 1–March 1 (regular years) and January 1–February 29 (leap years) due to the change of targeting area as suggested in the review's GENERAL COMMENT 2.

Please see lines 7-10 in page 3.

"In this study, we confined the analysis to the 60 coldest days of winter based on the climatologically averaged SST (January 1–March 1 in regular years, and January 1–February 29 in leap years; Supplementary Fig. 1). During these climatologically coldest days of winter (hereafter referred to as just "winter days"), further cooling days may be expected to have the greatest implications for aquaculture."

Figure S1

[Figure]

**Figure S1:** time series of climatological SST (black line) across the southern TS (the white dash rectangle in Figure 1) and critical temperature of cold water days (gray line). Red dash lines reveal the period of winter days.

5. *In addition, please confirm the definition of SSTA in line 17 of page 3. The sea surface temperature anomaly (SSTA) is the difference between the observed SST and the climatological SST. Did author use which the climatological daily SST is? In general, the SSTA is good indicator to see the long-term warming or cooling trend.*

**Reply:**

Yes. SSTA is a deviation from the daily climatological average (we will add a figure to clarify in Supplementary) and the time series of SSTA can be an indicator to study long-term trend. However, the trend in our studying area has a gentle slope (0.01°C/year) during the studying period of 1995-2017 (dark blue line in AC2-Figure 3). Actually, we can see a significant warming trend (0.21 °C/year) from 1995 to 2000 (red line), which is similar to the results of Kuo et al. (2017, Int. J. Remote. Sens.; 1980-2000 trend is about 0.15 °C/year) and Belkin et al. (2014, Clim. Change; 1978-1998 trend is about 0.07 °C/year). A trend has large variability depending on a sampling window, so you can't see an obvious long-term trend during the studying period. We have added a sentence for note in the revised manuscript.

Please see lines 21-22 in page 4.

[revised manuscript text omitted]